# Determining the influential factors of postnatal care in Bangladesh using multilevel logistic regression

**Imran Hossain Sumon, Md. Sifat Ar Salan, Mohammad Alamgir Kabir, Ajit Kumar Majumder, Md. Moyazzem Hossain** [ID]*

Department of Statistics and Data Science, Jahangirnagar University, Savar, Dhaka, Bangladesh

* hossainmm@juniv.edu

## Abstract

### Background

Postnatal care (PNC) is the care of a newborn and mother for up to six weeks from one hour of placenta birth. The postnatal period is one of the most hazardous stages for mothers and their baby's health. The PNC is influenced by several maternal, family, biological, and socio-economic factors and it is necessary to identify the most significant factors of PNC. Therefore, the authors focus on determining the significant determinants of postnatal care in Bangladesh.

### Methods and materials

This study is based on a secondary dataset extracted from the Bangladesh Demographic and Health Survey (BDHS)-2017/18. Descriptive statistics, chi-square test, and multilevel logistic regression have been used to determine the contributing factors of PNC.

### Results

The rate of postnatal care was highest in Sylhet (73.7%) and lowest in Dhaka (57.1%). Female babies had 10.1% less odds of having postnatal care than male babies. Findings depict that the children with 1–3 siblings have 1. 82 times more odds (odds ratio (OR):1.82, 95% confidence interval (CI):0.03–3.21) of PNC than babies without any siblings. Children who suffered from fever recently had 1.25 times (OR = 1.25, 95% CI:1.09–1.45) more odds of taking PNC than their counterparts. Children of working mothers had 1.33 times (OR:1.33, 95% CI:1.14–1.56) more odds of having PNC than children of non-working mothers.

### Conclusion

The sex of a child, birth order number, place of residence, region, receiving Bacillus Calmette-Guérin (BCG) vaccine, number of antenatal care visits, having fever recently, number of household members, media exposure, and household facilities are significantly linked with PNC in Bangladesh. To ensure the good health of a child, it is necessary to focus on the

**Data Availability Statement:** The data set used in this study will be available from the website The DHS Program. In order to gain access to the data

files, you have to complete the registration. The data set is available from the following link http://dhsprogram.com/data/available-datasets.cfm.

**Funding:** The author(s) received no specific funding for this work.

**Competing interests:** The authors have declared that no competing interests exist.

**Abbreviations:** BDHS, Bangladesh Demographic and Health Survey; OR, Odds ratio; CI, Confidence interval; PNC, Postnatal care; BCG, Bacillus Calmette-Guérin; SDGs, : Sustainable Development Goals; MICS, Multiple Indicator Cluster Survey; PPS, Probability proportional to size; BMI, Body mass index; AUC, Area under the curve.

targeted groups and put emphasis on the identified variables. The authors believe that the findings will be helpful to the policymakers of Bangladesh to lessen childhood morbidities which will be helpful in achieving the target of the Sustainable Development Goals (SDGs) for reducing preventable maternal and under-five deaths by 2030.

## Introduction

Postnatal care (PNC) is the care of a newborn and mother for up to six weeks (i.e., 42 days) from one hour of placenta birth [1]. The postnatal period is one of the most hazardous stages for mothers and their baby's health. Proper implementation of health care services during the vulnerable stage can reduce child and maternal mortality and morbidity (i.e., disabilities and injuries) to a great extent [2]. At this critical juncture, indifference to healthcare can be fatal for both mother and baby. The infant mortality rate in 1000 live births is 38 [3]. The risk of neonatal death is about four times that of the next 11 months [3]. The Bangladesh Multiple Indicator Cluster Survey (MICS)-2019 reported similar results, with a mortality of infant and neonatal out of 1,000 live births were 34 and 26 respectively [3]. Therefore, in order to lessen the adverse effects on the child and maternal health, proliferating attention has to be paid to the identification of postnatal care covariates. Despite worldwide accentuation on health care for children and pregnant women, the progression of infant and maternal mortality rates is falling very slowly. Researchers found that wages, education, media access, and health services had a significant effect on PNC and delivery services [4]. According to a survey of Indigenous peoples in Bangladesh, researchers pointed out that type of residence, age, proximity to health centers, education, and media access were significantly associated with antenatal care [5].

A research focusing on postnatal care considering BDHS-2014 data and highlighted that place of residence, region, resource indicator, prenatal care visits, sex of baby, baby size, age of women at 1st birth, educational status of the mother, awareness of a community clinic, order of birth and media access is involved in postnatal care [6]. A previous study on postnatal care considered the BDHS-2014 data and highlighted that place of birth, place of residence, region, religion, resource indicator, educational status, and family wealth index are important factors for postnatal care [7]. Living in an urban area and having a higher income were also strongly linked to PNC in Bangladesh [8, 9]. Post-natal care was shown to be much more likely to be received by women who watched television and significantly less likely by mothers with less education [9]. Additionally, postnatal care services—especially those provided by qualified practitioners in Bangladesh's rural areas—are far less likely to be used by displaced mothers for their newborns [10]. Researchers highlighted that educated mothers from wealthy families and urban areas participated in more than four antenatal care visits. Moreover, children are less likely to get PNC if their parents are involved in agriculture [11]. Several socioeconomic and demographic factors such as place of residence, maternal education, occupation of the husband, the number of previous pregnancies, regular visits for antenatal care, and wealth influence PNC reporting for newborns in Bangladesh [8, 12–19].

Socioeconomic and geographic inequalities in postnatal care exhibited substantial variation across components and among different states in India [20]. In Sub-Saharan African countries, the overall prevalence of PNC for a baby was low. Significant relationships were found between PNC and the respondents' age, education level, wealth status, ANC visits, place of delivery, housing, and community-level poverty [21]. It was discovered that postnatal care service utilization in Yemen was substantially correlated with the mode of delivery, the place of delivery,

and the receipt of postnatal care information during prenatal visits [22]. In Pakistan, it has been noted that there are higher odds for older women with emotional and decision-making autonomy to have caesarean section deliveries and to use PNCs for both the mother and the newborn [23]. Researchers observed that ANC was more effective in Asian countries, whilst PNC was more effective in African countries [24].

In Bangladesh, most of the studies consider the previous BDHS data and used the logistics regression approach to find the determinants of PNC. However, in the BDHS survey, stratified samples were taken in two stages where the first stage consists of geographical locations based on some common features, considered as clusters. Individuals are collected from each cluster in the second stage due to the fact that there is a probable correlation of individuals within the cluster. A multilevel model can be used for minimizing higher-order variations considering the BDHS data [25]. Group-level influence and the hierarchical structure of the data are revealed by Intra-class correlation (ICC), which measures the ratio of between-group variation to total variance in multilevel models. This information helps determine whether to use a multilevel framework [26, 27]. Because of its ability to quantify between-cluster variation and maintain non-independence, multilevel analysis offers benefits beyond the precise calculation of regression coefficients and standard errors. Furthermore, it is possible to allow the degree of variation in the relationship between the variables and the outcome between clusters, something that is difficult to achieve with conventional regression methods [28–31]. Consequently, the multilevel logistic model is incorporated to examine the important factors of postnatal care which can simultaneously include fixed effect covariate and cluster effects [26, 27]. The BDHS data have some between-cluster variation and in this situation, multilevel analysis provides the precise calculation of regression coefficients and standard errors. Moreover, there is a lack of studies based on multilevel modeling considering the BDHS-2017/18 dataset. Therefore, the authors want to check whether the multilevel modeling classifies the status of PNC more precisely or not than the logistic regression model. To fill up the existing gap in the literature and compare the classification accuracy, the authors aimed to identify the covariates that are affecting postnatal care using BDHS-2017/18 data and fit a suitable multilevel model and logistic regression model with identified potential covariates. The authors believed that the results will help Bangladeshi policymakers to lessen childhood morbidities, which will help them meet the Sustainable Development Goals (SDGs) aim of reducing preventable under-five child deaths by 2030.

## Materials and methods

### Data

This study is based on a secondary dataset extracted from the eighth national survey (*i.e.*, BDHS-2017-2018) to identify the influential factors of postnatal care (PNC). In this survey, a two-stage stratified sampling of households has been used; where 675 enumeration units were selected in the 1st stage by the probability proportional to size (PPS) sampling technique. Out of 675 units, 425 are in rural and 250 in urban. A complete family list was operated in all the enumeration units selected in the first stage to provide a sample frame in 2nd stage for the purpose of household selection. From each enumeration unit, 30 households were selected by using the systematic sampling procedure in 2nd stage. The survey was completed with 8,772 children in the remaining 672 clusters after eliminating a rural cluster in Rajshahi, a rural cluster in Rangpur, and an urban cluster in Dhaka due to natural disasters. After deleting the missing values, the final sample consisted of 4,729 children under 3 years of age. A weighted sample is used for the analysis to confirm the country's representative sample. The detailed sampling procedure and the guidelines for using weight are discussed in the report of BDHS-2017-2018 [3].

## Target variable

The target variable of this study was receiving postnatal care categorized as a binary variable which took only two values; "1" if the postnatal care is received and "0" for otherwise.

## Covariates

Socio-demographic and economic factors is considered in this as covariates including sex of children (male, female), age of child in month (<7, 7–24 and 24+), birth order number (1st, 2nd-3rd, 4 or more), receiving BCG (No, Yes) vaccine, death of any child (No, Yes), number of siblings (None, 1–3, 4 or more), had fever recently (Yes, No), number of antenatal care visit (None, 1–3, 4 or more), number of entry in pregnancy and PNC roaster (1, 2 or more), body mass index (thin, normal, overweight, obesity), age of mother (<18, 18–25, 26–35, 36 or more years), age of mother at marriage (<16, 16–21, 22 or more years), age of mother at first birth (<16, 16–21, 22 or more years), currently working status of mother (No, Yes), mothers education level (no education and primary, secondary, higher than secondary), fathers education level (no education and primary, secondary, and higher than secondary), fathers occupation (job, business, others), wealth index (poorer, poorest, middle, richer, richest), media exposure (No, Yes), number of household member (<4, 4–6, 7 or more), toilet facility (flash toilet, pit toilet, others), and cooking fuel (gas/electricity, wood, others), region (Dhaka, Chattogram, Barishal, Khulna, Sylhet, Mymensingh, Rangpur and Rajshahi), type of residence (rural, urban), religion (Islam, others). The choice of variables was influenced by the availability in the BDHS-2017/18 dataset, self-efficacy, as well as relevant literature [5, 6, 8–10, 12–18].

## Statistical methods

Bivariate analysis was performed in this study to find out the socio-demographic covariates that affect postnatal care. Pearson chi-square test was used to identify the covariates for the logistic regression model and whether they were significantly associated with postnatal care or not. Sankey plot is used to visualize the relationship between the target variable and selected covariates [32]. With the help of multi-level logistic regression, significantly contributing factors of postnatal care in Bangladesh have been identified. The BDHS-2017/18 dataset follows a hierarchical structure. Individuals are considered as the level-1 and clusters as the level-2 for this study hierarchy. Silvia introduces the applications of multilevel modeling for nested and hierarchical data [33]. The matrix form of the two-level logistic regression is as follows:

**Level I:**

$$\ln\left(\frac{P_j}{1 - P_j}\right) = \beta_j^T X_j$$

$$P_j = \frac{\exp(\beta_j^T X_j)}{1 + \exp(\beta_j^T X_j)}; j = 1, 2, \ldots, m$$

where, $\beta_j^T = (\beta_{0j}, \beta_{1j}, \ldots, \beta_{pj})$, $X_j = (1, X_{1j}, X_{2j}, \ldots, X_{pj})$, and $P_j = (P_{1j}, P_{2j}, \ldots, P_{nj})$; $P$ is the number of independent variables with a variable for which intercept is present, $n$ is the total number of observations in a cluster, $m$ is the number of total clusters and $P_{ij}$ represents the probability of the $i^{th}$ individual in the $j^{th}$ level-2 unit [34].

**Level II:**

$\beta_{0j} = \gamma_{00} + u_{0j}$ and $\beta_{1j} = \gamma_{10}$.

The model has two components, $u_{0j} \sim N(0, \tau_{00})$ and $\varepsilon_j \sim N(0, \sigma^2)$. Therefore, the intra-cluster correlation is defined as $\frac{\sigma^2}{\sigma^2 + \tau_{00}}$ [34].

In multilevel analysis, strong assumptions to perform multilevel logistic regression: (i) random effects are normal (or, when the slope is as long as random intercepts, that the joint distribution is multivariate normal), (ii) all of the relevant variables should be included in the model, so that the researchers are safe assuming that regressors and errors are uncorrelated in all levels, (iii) the author should have enough observations at every level to really utilize the results of asymptotic theory concerning the about the test statistics of likelihood ratio and the inverse of the fisher information matrix as the estimator of the variances of the parameter estimates. The residuals of level-1 are supposed to follow normality, and the residuals of level-2 should follow multivariate normality, moreover, the residuals of level-2 and level-1 are independent. Formally, these assumptions in matrix notation are expressed as:

$$E \begin{pmatrix} u \\ \varepsilon \end{pmatrix} = \begin{pmatrix} 0 \\ 0 \end{pmatrix} \text{ and } Var \begin{pmatrix} u \\ \varepsilon \end{pmatrix} = \begin{pmatrix} G & 0 \\ 0 & R \end{pmatrix}$$

where, $G$ represents the covariance matrix for residuals of level-2, and $R$ represents the covariance matrix for residuals of level-1.

## Ethical approval

This study considers a secondary publicly available data set. It is not required as the initial survey was approved by the Ethics Committee of the ICF Macro at Calverton in the USA and by the Ethics Committee in Bangladesh. This study is based on the secondary data that is extracted from the Bangladesh Demographic and Health Survey- 2017/18 and the dataset used in it contained no personally identifiable information about the survey respondents. The initial survey was taken proper consent from the participants and followed the ethical standard.

## Results

The percentage distribution of PNC by different background characteristics is given in Table 1. Findings revealed that the sex of the child, receiving the BCG vaccine, having fevers of the child recently, pregnancy, and PNC care roaster entry are significantly associated with PNC receiving. Male children have a higher rate (66.5%) of postnatal care than female children (62.8%). The birth order is positively associated with the postnatal care rate. Children with a recent fever were more likely to have postnatal care. Children who receive BCG vaccine have a higher prevalence (65.2%) of postnatal care than children who do not receive BCG (58.4%) [Table 1].

Although child age is an important determinant, unfortunately, no significant association between the age of children and postnatal care is observed in this study. Results also disclosed that parental characteristics like the age of women at the time of marriage, the age of the mother at first birth, the mother's educational level as well as occupational status, the father's educational status, and occupation are significantly associated with PNC receiving. The younger married women and the duration of first birth are linked with the rate of postpartum care. The higher the father's educational qualifications decreased the postnatal care rate. Table 1 reveals that mothers who are working have a postnatal care rate of 70.8% and those who are not working have a postnatal care rate of 61.2%. The rate of postnatal care is highest in Sylhet (73.7%) and lowest in Dhaka (57.1%). The prevalence of postnatal care in urban and rural areas is 61.5% and 65.9% respectively. Postnatal care of a child in a Muslim family is higher

**Table 1. Percentage distribution of selected covariates with the status of postnatal care.**

| Characteristics | Postnatal care (Yes) | | Characteristics | Postnatal care (Yes) | |
|---|---|---|---|---|---|
| | *n* | % | | *n* | % |
| **Sex of child**** | | | **Mother's education level**\*** | | |
| Male | 1650 | 66.5 | No education | 207 | 71.1 |
| Female | 1411 | 62.8 | Primary | 921 | 71.2 |
| **Age of child in months** NS | | | Secondary | 1467 | 63.0 |
| < 7 | 389 | 64.1 | Higher | 466 | 57.1 |
| 7–24 | 933 | 65.7 | **Father's Education level**\*** | | |
| 25 or more | 1714 | 63.4 | No education | 488 | 73.9 |
| **Birth order number**\*** | | | Primary | 1090 | 68.5 |
| 1 | 1102 | 61.6 | Secondary | 983 | 61.4 |
| 2–3 | 1535 | 64.9 | Higher | 500 | 57.1 |
| 4 or more | 424 | 74.0 | **Father's occupation**\*** | | |
| **Receiving BCG*** | | | Job | 618 | 59.9 |
| No | 192 | 58.4 | Business | 524 | 60.9 |
| Yes | 2870 | 65.2 | Others | 1920 | 67.7 |
| **Death of any child*** | | | **Wealth index combined**\*** | | |
| No | 2747 | 64.2 | Poorest | 745 | 75.3 |
| Yes | 314 | 69.3 | Poorer | 677 | 70.2 |
| **Number of siblings**\*** | | | Middle | 578 | 63.0 |
| None | 1157 | 61.4 | Richer | 547 | 57.0 |
| 1–3 | 1576 | 65.4 | Richest | 514 | 57.2 |
| 4 or more | 328 | 75.4 | **Media exposure**\*** | | |
| **Had fever recently**** | | | Yes | 1794 | 70.0 |
| No | 1886 | 63.2 | No | 1268 | 58.5 |
| Yes | 1175 | 67.5 | **Number of Household member*** | | |
| **Number of ANC visit**\*** | | | < 3 | 331 | 59.6 |
| No visit | 290 | 76.7 | 4–6 | 1700 | 64.8 |
| 1–3 | 1360 | 64.7 | 7 or more | 1030 | 66.4 |
| 4 or more | 1411 | 62.8 | **Toilet Facility**\*** | | |
| **Pregnancy and PNC roaster entry*** | | | Flash toilet | 682 | 57.3 |
| 1 | 2881 | 64.4 | Pit toilet | 1920 | 68.3 |
| 2 or more | 180 | 70.9 | Others | 459 | 63.1 |
| **Body mass index**** | | | **Cooking fuel**\*** | | |
| Thin | 484 | 66.9 | Gas/Electricity | 492 | 58.9 |
| Normal | 1940 | 65.9 | Wood | 2163 | 66.8 |
| Overweight | 515 | 60.6 | others | 406 | 61.8 |
| Obesity | 122 | 57.3 | **Region**\*** | | |
| **Age of Mother** NS | | | Barisal | 191 | 70.5 |
| < 18 | 82 | 63.7 | Chittagong | 686 | 68.5 |
| 18–25 | 1550 | 63.8 | Dhaka | 670 | 57.1 |
| 26–35 | 1247 | 66.1 | Khulna | 272 | 63.4 |
| 36 or more | 186 | 65.6 | Mymensingh | 293 | 70.8 |
| **Age at marriage**** | | | Rajshahi | 330 | 58.7 |
| < 16 | 1080 | 67.1 | Rangpur | 351 | 68.2 |
| 16–21 | 1829 | 64.1 | Sylhet | 269 | 73.7 |
| 22 or more | 153 | 57.1 | **Type of residence**** | | |
| **Age at first birth**** | | | Urban | 756 | 61.5 |

*(Continued)*

**Table 1.** (Continued)

| Characteristics | Postnatal care (Yes) | | Characteristics | Postnatal care (Yes) | |
|---|---|---|---|---|---|
| | *n* | % | | *n* | % |
| < 16 | 468 | 67.7 | Rural | 2305 | 65.9 |
| 16–21 | 1749 | 66.0 | **Religion**** | | |
| 22 or more | 844 | 60.7 | Islam | 2841 | 65.3 |
| **Mother currently working**\*\*\* | | | Others | 221 | 58.2 |
| No | 1822 | 61.2 | | | |
| Yes | 1239 | 70.8 | | | |

**Note:** NS: Not significant

*: p-value <0.05

**: p-value <0.01

***: p-value <0.001

than in other religious families. Table 1 exhibits that the rate of postnatal care was higher if a child died before it [Table 1].

Fig 1 illustrates the association among the sex of the child, the mother's education status, and the wealth index with the status of PNC. It is observed that approximately 48% of children's mother education level is secondary, and only 6.1% of mothers have no education. More mothers who have higher education were from the richest families [Fig 1].

Furthermore, findings revealed that the difference between clusters is noticeable because the variance is 1.91. The value of intra-cluster correlation is 0.367 which indicates that 36.7% of the total variation in postnatal care happened because of the discrepancies between the clusters.

In Fig 2, the green line narrates that the fitted model had no efficiency in classifying PNC acceptance (i.e., "no data classifier"). True positive rates and false positive rates of fitted models of multilevel logistic regression and logistic regression for different probabilities or marginal values represented by red and blue lines, respectively. Findings reveal that the AUC of multilevel logistics regression is 0.771 which points out that about 77% of predictions were done correctly and the value of AUC for the logistics regression model is 0.636 which indicates that almost 64 out of 100 predictions were done accurately. Therefore, multilevel (i.e., nested) logistic regression provides 13% more accurate predictions than classical logistic regression [Fig 2].

Table 2 shows that among own factors of a child; sex of a child, having fevers recently, birth order, BCG receiving, pregnancy, and care roaster entry of a child were significantly linked with the status of PNC. The number of antenatal care visits produced a significant influence on PNC receiving. Female babies have a 10.1% lower odds of receiving postnatal care than male babies. Children who suffered from fever recently had 1.25 times (OR: 1.25, 95% CI: 1.09–1.45) more odds of taking PNC than their counterparts. The relationship between postnatal care and birth order is reversed, meaning that children with a higher order of birth were less likely to have postnatal care. Babies with birth order 2nd or 3rd have 0.54 times (OR: 0.54, 95% CI: 0.30–0.96) odds of receiving PNC than the first child. Furthermore, babies of order 4th or above had 0.45 times odds of receiving PNC (OR: 0.45, 95% CI: 0.18–1.13) than the first child. Babies who received the BCG vaccine had 1.43 times (OR: 1.43, 95% CI: 1.10–1.86) more odds of having PNC than those who did not receive the BCG vaccination [Table 2].

Among parental factors; the father's educational qualification, the mother's occupational status, and the number of antenatal care visits have a significant influence on PNC receiving. Children of working mothers had 1.33 times (OR:1.33, 95% CI:1.14–1.56) more odds of having

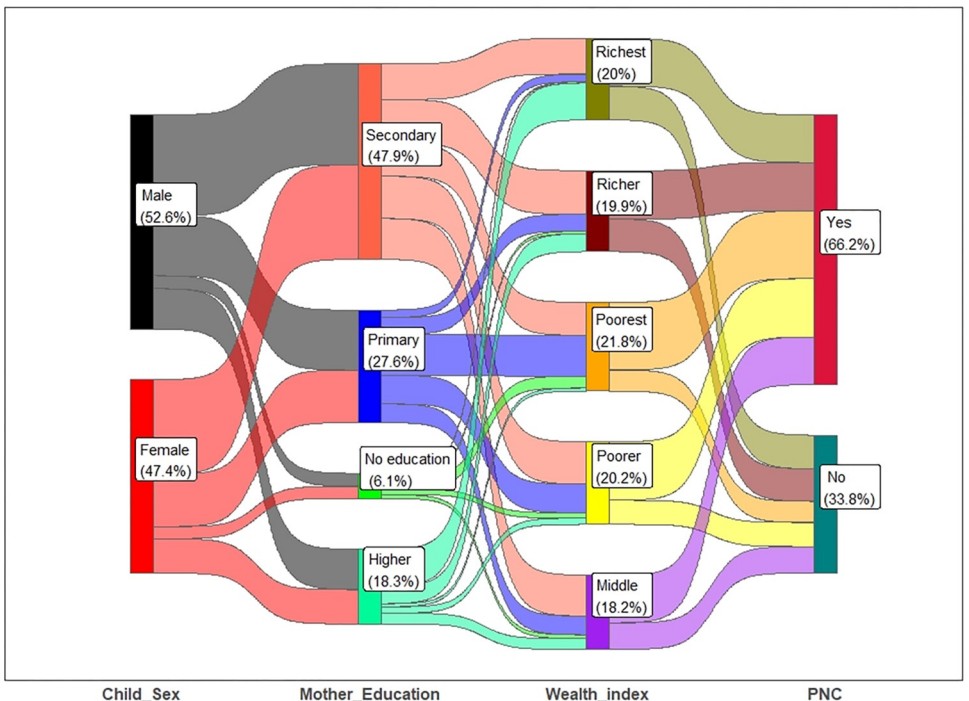

**Fig 1. Association among sex of the child, mother's education, wealth index with PNC.**

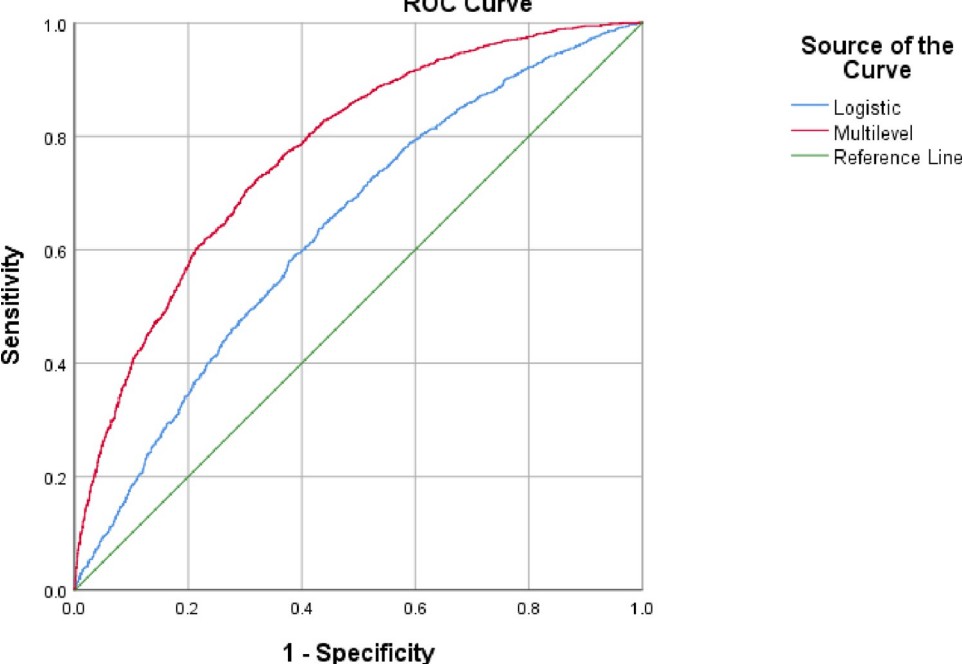

**Fig 2. The area under the curve (AUC) of the fitted models for the multilevel logistic and logistic model.**

**Table 2. Regression coefficients, odds ratios (OR), and 95% CI of odds ratios (OR) of multilevel logistic regression for postnatal care (PNC) in Bangladesh.**

| Characteristics | Coefficient | OR | 95% CI of OR | Characteristics | Coefficient | OR | 95% CI of OR |
|---|---|---|---|---|---|---|---|
| **Constant** | 1.040*** | 2.830 | [1.39, 5.74] | **Father's education level** | | | |
| **Sex of child** | | | | No education (Ref.) | - | - | - |
| Male (Ref.) | - | - | - | Primary | -0.160 | 0.852 | [0.68, 1.07] |
| Female | -0.118* | 0.899 | [0.78, 1.01] | Secondary | -0.298** | 0.742 | [0.58, 0.95] |
| **Birth order number** | | | | Higher | -0.376** | 0.686 | [0.51, 0.93] |
| 1 (Ref.) | - | - | - | **Father's occupation** | | | |
| 2–3 | -0.620** | 0.538 | [0.30, 0.96] | Job (Ref.) | - | - | - |
| 4 or more | -0.799* | 0.450 | [0.18, 1.13] | Business | -0.057 | 0.945 | [0.77, 1.16] |
| **Receiving BCG** | | | | Others | 0.124 | 1.131 | [0.95, 1.35] |
| No (Ref.) | - | - | - | **Wealth index combined** | | | |
| Yes | 0.358*** | 1.431 | [1.10, 1.86] | Poorest (Ref.) | - | - | - |
| **Death of any child** | | | | Poorer | 0.020 | 1.020 | [0.81, 1.29] |
| No (Ref.) | - | - | - | Middle | -0.111 | 0.895 | [0.70, 1.15] |
| Yes | 0.097 | 1.102 | [0.77, 1.57] | Richer | -0.197 | 0.821 | [0.62, 1.08] |
| **Number of siblings** | | | | Richest | -0.029 | 0.972 | [0.68, 1.39] |
| None (Ref.) | - | - | - | **Media exposure** | | | |
| Between 1–3 | 0.600** | 1.822 | [1.03, 3.21] | Yes (Ref.) | - | - | - |
| Above 3 | 0.927** | 2.527 | [1.04, 6.15] | No | -0.224*** | 0.799 | [0.68,0.95] |
| **Had fever recently** | | | | **Number of household member** | | | |
| No (Ref.) | - | - | - | < 4 | - | - | - |
| Yes | 0.226*** | 1.253 | [1.09, 1.45] | 4–6 | 0.163 | 1.177 | [0.94, 1.47] |
| **Number of ANC visit** | | | | 7 or more | 0.270** | 1.310 | [1.03, 1.67] |
| No visit (Ref.) | - | - | - | **Toilet facility** | | | |
| 1–3 | -0.311** | 0.733 | [0.55, 0.97] | Flash toilet (Ref.) | - | - | - |
| 4 or more | -0.151 | 0.860 | [0.64, 1.15] | Pit toilet | 0.184* | 1.202 | [0.97, 1.49] |
| **Pregnancy and PNC roaster entry** | | | | Others | 0.279 | 1.322 | [0.73, 2.41] |
| 1 (Ref.) | - | - | - | **Cooking fuel** | | | |
| 2 or more | 0.310* | 1.364 | [0.99, 1.88] | Gas/Electricity (Re.) | - | - | - |
| **Body mass index** | | | | Wood | -0.239* | 0.787 | [0.61, 1.02] |
| Thin (Ref.) | - | - | - | Others | -0.493 | 0.611 | [0.32, 1.16] |
| Normal | -0.004 | 0.996 | [0.82, 1.21] | **Region** | | | |
| Overweight | -0.018 | 0.982 | [0.78, 1.24] | Barisal (Ref.) | - | - | - |
| Obesity | -0.060 | 0.942 | [0.65, 1.36] | Chittagong | -0.018 | 0.982 | [0.66, 1.47] |
| **Age at marriage (years)** | | | | Dhaka | -0.579*** | 0.560 | [0.37, 0.84] |
| < 16 (Ref.) | - | - | - | Khulna | -0.367 | 0.693 | [0.45, 1.08] |
| 16–21 | -0.060 | 0.942 | [0.78, 1.13] | Mymensingh | -0.080 | 0.923 | [0.57, 1.49] |
| 22 or more | -0.192 | 0.825 | [0.58, 1.18] | Rajshahi | -0.634*** | 0.530 | [0.34, 0.82] |
| **Age at first birth (years)** | | | | Rangpur | -0.211 | 0.810 | [0.52, 1.27] |
| < 16 (Ref.) | - | - | - | Sylhet | 0.085 | 1.089 | [0.69, 1.73] |
| 16–21 | 0.077 | 1.080 | [0.85, 1.37] | **Type of residence** | | | |
| 22 or more | -0.025 | 0.975 | [0.74, 1.29] | Urban (Ref.) | - | - | - |
| **Mother currently working** | | | | Rural | -0.114 | 0.892 | [0.69, 1.15] |
| No (Ref.) | - | - | - | **Religion** | | | |
| Yes | 0.287*** | 1.332 | [1.14, 1.56] | Islam (Ref.) | - | - | - |
| **Mother's education level** | | | | Others | -0.291** | 0.748 | [0.56, 0.99] |
| No education (Ref.) | - | - | - | **Variance** | **1.907** | | |
| Primary | 0.074 | 1.076 | [0.74, 1.56] | **Correlation** | **0.367** | | |

(*Continued*)

**Table 2.** (Continued)

| Characteristics | Coefficient | OR | 95% CI of OR | Characteristics | Coefficient | OR | 95% CI of OR |
|---|---|---|---|---|---|---|---|
| Secondary | -0.020 | 0.980 | [0.71, 1.35] | | | | |
| Higher | -0.002 | 0.998 | [0.68, 1.48] | | | | |

**Note:** Ref.: Reference category

*: p-value <0.10

**: p-value <0.05

***: p-value <0.01

PNC than children of non-working mothers. It is obvious that the relationship between postnatal care and the father's educational qualification is reversed, meaning that in spite of increasing father's educational qualifications, the child's postnatal care is declining. A child of a father who has completed primary, secondary, and higher secondary levels has approximately 15% (OR = 0.85, 95% CI = 0.68–1.07), 26% (OR = 0.74, 95% CI:0.58–0.95), and 31% (OR = 0.69, 95% CI = 0.51–0.93) fewer odds to receive PNC respectively than a child whose father was illiterate. In the case of receiving PNC, no significant difference is observed between the different age groups of women, at the time of marriage, the age of the mother at first birth, the mother's educational qualification, and the father's occupation [Table 2].

The findings presented in Table 2 show that among the household character; the number of siblings of a child, religion, and media exposure are important determinants of PNC. It is conspicuous that the number of siblings increased the prevalence of postnatal care. Babies with 1–3 siblings had 1.82 times (OR: 1.82, 95% CI: 0.03–3.21) more odds of having PNC than babies without any siblings. Children of other religions had less odds of having postnatal care than Islam. Children in families without media exposure had 0.80 times (OR: 0.80, 95% CI: 0.68–0.95) less odds of receiving postnatal care than children in families with media exposure. Moreover, the region, number of family members, toilet facilities, and cooking materials are significantly associated with PNC receiving. A child born in the Sylhet division is more likely to get postnatal care than a child born in the Barisal division. A child in the Rajshahi division compared to the Barisal division had 0.53 times (OR: 0.53, 95% CI: 0.34–0.82) lower odds of getting postnatal care. Moreover, a family with more than 4–6 members had approximately 1.18 times (OR: 1.18, 95% CI: 0.94–1.47) odds of having postnatal care compared to a family having less than 4 members. Table 2 revealed that children with more than 3 siblings have nearly 2.53 times (OR: 2.53, 95% CI: 1.04–6.15) higher odds of receiving postnatal care than children without siblings. Wealth index and place of residence were found to be insignificant, i.e., there were no significant differences in PNC taking by wealth index, and place of residence [Table 2].

## Discussion

The authors aimed to identify the determinants of PNC in Bangladesh because the postnatal period is considered to be the most important and intricate time in the life of mothers and babies, but it is still the most neglected in developing countries like Bangladesh. A little carelessness at this time can lead to the death or disability of the newborn or mother. Postnatal care is a knot of interventions desired for the mother's and infant's health to medicate and prevent complications. Findings revealed that postnatal care for the second and third child and also for the fourth and more children are significantly less chance to have postnatal care than the first child. One of the reasons behind this may be that at higher birth order, women gain confidence and full knowledge about the child as well as maternal health. Preceding studies

revealed that maternity services are more likely to be used during the birth of the first child due to the risk of the first pregnancy [6, 35, 36]. This study showed the significant relationship between postnatal care with having a fever of babies recently and receiving the BCG vaccine. A previous study agreed with this statement [6]. However, the study exhibits an insignificant relationship between the age of the mother at first birth and postnatal care. This is not expected but similar to the previous work [6]. This study shows a significant but adverse effect of father's education on PNC which is contradictory to the previous study [37]. Although the age of the child and the current age of the mother appear to be important determinants of postnatal care, the survey did not show a statistical relationship between these two determinants of postpartum care, and several studies have shown similar results [6, 37]. This study showed the significant effect of the current working status of the mother and the education of the father on postnatal care and a previous study partially agrees with this statement [6]. Prior studies found a significant relationship between PNC visits and education for mothers [9, 38]. Children with highly educated parents are aware of the importance of child health and take the appropriate action [9, 18, 39–45].

Children in families without media exposure are less likely to have postnatal care than children in families with media exposure. A study highlighted that compared to women who do not watch TV, those who did had a considerably increased chance of acquiring PNC. Only the PNC found reading newspapers and magazines to be significant [9]. One of the reasons for this is that the media discusses the importance of postnatal care as well as raising awareness about child and maternal health. Women can increase their knowledge and awareness through using different media [46]. This finding is also shown in the previous study [6, 7, 47]. Disparities in postnatal care provision are exhibited between different geographical regions within a country. Differences in healthcare access, infrastructure, economics, culture, and government policies may be the reasons that lead to geographic disparities in postnatal care. A previous study revealed the same findings [7]. Compared to a child from a Muslim family, a child born into a Hindu or other religious family is likely to receive less postnatal care this finding is consistent with others [9]. There could be a unique cultural or religious practice or custom that contributes to potential differences in postnatal care between children from Muslim and Hindu or other religious families. A previous study revealed the same findings [7]. No significant inequality was revealed in the acceptance of postnatal care between urban and rural mothers. This is an expected result but contradictory to the previous study [6, 47]. One of the reasons for this may be the increase in the rate of education in rural areas. Other reasons may be that access to healthcare services, media access is increasing in rural areas.

In order to improve the health of the child and maternal health by raising awareness on postnatal care of the mother and the family, there should be a massive campaign in the media. Emphasis should be placed on making media access easily available in rural areas as well as at the same time providing newspapers. In order to get the expected results, it is necessary to facilitate media access in rural areas as well as to provide newspapers at the same time. However, the government should take necessary steps to raise awareness among women and their families about maternal and child health care after delivery so that adverse effects can be avoided by adopting new strategies.

## Strengths and limitations

The strength of this study is the originality and based on the most recent country-representative dataset. The influential determinants were determined by the multilevel logistic regression model. While our results provide significant introspection into the determinant of postnatal care, they need to be interpreted with the consideration of analysis limitations. Although there

may be regional and temporal variation in the prevalence of the PNC, the authors do not include it in this study; however, it is a potential topic for future research. Moreover, in a future study, relevant determinants of postnatal care in Bangladesh can be marked out using the machine learning classifier method within the multilevel logistics model. Due to the higher inclusion of this study model compared to the previous studies, some differences have been observed in the analysis.

## Conclusion

The receiver operating curve reveals that the performance of the multilevel logistic regression model is better than that of the classical logistics model i.e., multilevel logistic regression provides 12% more accurate predictions than classical logistic regression. This study's findings revealed that region, type of residence, child's sex, BCG vaccination, birth order number, number of siblings, recent fever, number of antenatal care visits, pregnancy, and care roaster entries, mother's current employment status, Father's educational status, media exposure, number of household members, toilet facility, and cooking fuel were all found to be strongly linked to postnatal care. To ensure the good health of a child, it is necessary to focus on the targeted groups and put emphasis on the identified variables. Mothers should have access to more health education, including transportation and counseling to address various dangers and promote awareness. The existing maternity and child health facilities should place a high priority on providing proper postnatal care. The authors believed that the findings will be beneficial to the policymakers of Bangladesh to lessen childhood morbidities which will be helpful to achieve the target of the Sustainable Development Goals (SDGs) for lessening preventable under-five child deaths by 2030.

## Acknowledgments

The authors are grateful to ICF International, Rockville, Maryland, USA, for providing the Bangladesh DHS datasets for this analysis. The authors also thankful to the anonymous reviewers and academic editors for their valuable comments and feedback that helps to improve the quality of the manuscript.

## Author Contributions

**Conceptualization:** Imran Hossain Sumon, Md. Sifat Ar Salan, Mohammad Alamgir Kabir, Ajit Kumar Majumder, Md. Moyazzem Hossain.

**Data curation:** Md. Moyazzem Hossain.

**Formal analysis:** Imran Hossain Sumon, Md. Sifat Ar Salan.

**Methodology:** Ajit Kumar Majumder, Md. Moyazzem Hossain.

**Supervision:** Mohammad Alamgir Kabir, Ajit Kumar Majumder.

**Validation:** Mohammad Alamgir Kabir, Md. Moyazzem Hossain.

**Visualization:** Imran Hossain Sumon, Md. Moyazzem Hossain.

**Writing – original draft:** Imran Hossain Sumon, Md. Sifat Ar Salan, Md. Moyazzem Hossain.

**Writing – review & editing:** Mohammad Alamgir Kabir, Ajit Kumar Majumder, Md. Moyazzem Hossain.

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
