## [Decision Letter · Decision Letter 0]

3 Nov 2023

PONE-D-23-08999Determining the Influential Factors of Postnatal Care in Bangladesh using Multilevel Logistic RegressionPLOS ONE

Dear Dr. Hossain, 

Thank you for submitting your manuscript to PLOS ONE. After careful consideration, we feel that it has merit but does not fully meet PLOS ONE’s publication criteria as it currently stands. Therefore, we invite you to submit a revised version of the manuscript that addresses the points raised during the review process.

We look forward to receiving your revised manuscript.

Kind regards,

Shafiun N. Shimul, PhD

Academic Editor

PLOS ONE

Journal Requirements:

3.Please include captions for your Supporting Information files at the end of your manuscript, and update any in-text citations to match accordingly. Please see our Supporting Information guidelines for more information: http://journals.plos.org/plosone/s/supporting-information.   

Reviewers' comments:

Reviewer's Responses to Questions

**Comments to the Author**

1. Is the manuscript technically sound, and do the data support the conclusions?

Reviewer #1: Partly

Reviewer #2: Yes

2. Has the statistical analysis been performed appropriately and rigorously? 

Reviewer #1: Yes

Reviewer #2: Yes

3. Have the authors made all data underlying the findings in their manuscript fully available?

Reviewer #1: Yes

Reviewer #2: Yes

4. Is the manuscript presented in an intelligible fashion and written in standard English?

Reviewer #1: No

Reviewer #2: Yes

5. Review Comments to the Author

Reviewer #1: Comments to the authors:

General comments:

The study is lacking novelty with absence of rationality, knowledge gap, and unsupported conclusions with findings. The authors should revise the introduction section to clearly outline the novelty and significance of their study. This should include a comprehensive rationale and a well-defined knowledge gap that their research aims to fill. The authors should carefully reassess their findings, ensuring that their conclusions are fully supported by the presented data. It is important to address any discrepancies and provide a more rigorous analysis and interpretation of the results. Additionally, I suggest that the authors strengthen the discussion section by highlighting the previous studies in different settings, and implications of the findings in order to enhance the overall quality of the manuscript.

Specific comments:

Abstract

- It lacks rationale for determining the contributing factors of postnatal care. Authors should add this.

- Objective does not support the conclusions.

- What is your suggestion based on study findings?

Introduction:

- This section is severely disorganized.

- There is absence of significance of the study

- \\Knowledge gap of the study is ;[not clear

- The authors stated that "However, most of the previous studies used logistic regression for finding the contributing factors of PNC. Because of the correlation, it is obligatory to consider cluster effect on regression models when performing postpartum care analysis that will yield efficient results than the general logistic regression model." What is the basis for claiming "Because of the correlation, it is obligatory to consider cluster effect on regression models when performing postpartum care analysis that will yield efficient results than the general logistic regression model"?. It is there any theoretical background or empirical evidence?

- This section is lacking the literature related to determinants of PNC in Bangladesh and other developing countries and the reflection of receiving PNC helps to lessen childhood mortalities which is claimed by the authors.

- What kind of strategy was adopted and launched by the government and public in Bangladesh and other developing countries with similar circumstances to improve the rate of receiving PNC? What are the factors they emphasized for influencing mothers to take PNC?

- What are the benefits to build society and nation of the study? Not stated.

Methodology:

- Replace “Methodology” with Materials and Methods

- The authors claimed as follows "the choice of variables was influenced by the availability in the BDHS-2017/18 dataset, self-efficacy, as well as relevant literature." What are the literatures?

- What are the criteria for selecting the variables? Did you perform any customization of the variables? If yes, based on what?

Results:

- Move the following sentence in the methods section as it represents method of significance test "Pearson chi-square test was used to identify the covariates whether they were significantly associated with postnatal care or not."

- Sometimes percentage used as %, sometimes as percent. Need to follow uniformity of presentation

- Findings not presented maintaining coherence and cohesion

- What are new in your findings? Are there any new findings than those of in existing literature? Does the result produced by your regression model have similarity or dissimilarity with the findings from logistic regression or any other regression model?

- This section is so redundant as many parts of the tables mentioned in text of this section.

Discussion:

- Start your discussion with providing the answer to your research question

- This section missed some important findings of the study. Need to present only the crucial findings with enough implications of the findings which are lacking

- Literatures that support the study findings are not well articulated for the readers i.e. which settings, what is the actual findings??

- Though study findings showed there some important factors influencing PNC receiving, the authors emphasized on only media campaign, newspaper. Why? Newspaper, how? Is newspaper campaign supported by your findings?

Conclusion:

- This section is also confusing and did not reflect the actual work done

- I think first six sentences of this section are irrelevant here

- The authors stated that they believed that the findings will be beneficial to the policymakers of Bangladesh to lessen the childhood morbidities which will be helpful to achieve the target of the Sustainable Development Goals (SDGs) for lessening preventable maternal as well as under-five deaths by 2030. How? What are your suggestions based on your findings? These are absent.

Overall: Language editorial is extensively required. I believe that with appropriate revisions and addressing the concerns raised, this manuscript has the potential to make a valuable contribution to the field. I would recommend inviting the authors to revise and resubmit their work, giving them an opportunity to address the issues raised in this review.

Reviewer #2: The study aims to examine important factors associated with postnatal care in Bangladesh using a secondary dataset extracted from the Bangladesh Demographic and Health Survey 2017-18. Multiple and Multilevel logistic regressions have been used to determine the key factors related to postnatal care. The results are important for policy makers.

The manuscript is systematically written with logical arguments and the evidence are solid to claim the statements.

6. PLOS authors have the option to publish the peer review history of their article (what does this mean?). If published, this will include your full peer review and any attached files.

Reviewer #1: **Yes: **Md. Ragaul Azim

Reviewer #2: No

---

## [Author Response · Author response to Decision Letter 0]

22 Nov 2023

Dear Editor,

We would like to sincerely thank the anonymous reviewers, the Academic Editor, for their valuable comments. We have considered all comments and then thoroughly revised and formatted the manuscript. A detailed response to each comment is provided below.

Author's Response to Editor comments:

1. Thanks. We appreciate your feedback. As per comments, a careful revision has been conducted, and all required files are uploaded to the journal submission system. The revised texts are highlighted in “red” color.

Author's Response to Journal Requirements:

1. Thanks. We revised the format of the manuscript following the PLOS ONE style. 

The revised texts are in “red”. Page: 1-16

2. Thanks. We add the Ethics Statement in the manuscript and submission system. 

The revised texts are in “red”. Page: 7

3. Thanks. We have no supplementary files for this manuscript. 

Author's Response to Reviewer 1 comments:

1. Thank you very much for carefully checking the manuscript and providing comments and feedback. We believe that this helps to improve the quality of the manuscript. 

2. Thanks. We revised the Abstract section as per your comments. 

The revised texts are in “red”. Page: 1-2

3. Thank you very much for your comments. We revised the Introduction section. 

The revised texts are in “red”. Page: 2-4

4. Thanks. We change it as per your suggestion. We revised this section. 

The revised texts are in “red”. Page: 4-7

5. Thank you very much. We moved it to the Methods and Materials section. We also check % and percent and maintain uniformity. 

We revised the whole Result section. 

The revised texts are in “red”. Page: 7-13

6. Thanks. We revised the Discussion section. 

The revised texts are in “red”. Page: 13-15

7. Thanks for your suggestion. We have revised the manuscript. 

The revised texts are in “red”. Page: 15-16

8. We appreciate this comment. The Methods section has been revised. 

The revised texts are in “red”. Page: 1-16

Author's Response to Reviewer 2 comments:

The authors are thankful to the reviewer for the motivational comments. It motivates young researchers to do research from low-income countries like Bangladesh without any funding support. 

In conclusion, the revised version of the manuscript has been produced as per the review outcomes. So, we hope that you will be happy to see this greatly improved version. Once again, we would like to thank you all for your dedication, professional services, and cooperation.

---

## [Decision Letter · Decision Letter 1]

6 Sep 2024

PONE-D-23-08999R1Determining the Influential Factors of Postnatal Care in Bangladesh using Multilevel Logistic Regression

PLOS ONE

Dear Dr. Hossain,

Thank you for submitting your manuscript to PLOS ONE. After careful consideration, we feel that it has merit but does not fully meet PLOS ONE’s publication criteria as it currently stands. Therefore, we invite you to submit a revised version of the manuscript that addresses the points raised during the review process.

We look forward to receiving your revised manuscript.

Kind regards,

Mohammed Moinuddin, PhD

Academic Editor

PLOS ONE

Journal Requirements:

Additional Editor Comments:

**Comment 1 PNC definition – **while reading the paper it feels sometimes that the PNC for newborn and sometime PNC for both mothers and newborn. Please check and make clear, I could be wrong though.

**Comment 2** **Interpretation issue** – authors interpretation of odds ratio in the fashion “Children of working mothers had 33 percent (OR:1.33, 95% CI:1.14-1.56) more likelihood to have PNC than children of non-working mothers.” is misleading.

Odds is not a measure of ‘likeliness’ or ‘chance’ rather it is a ratio of number of successes divided by the number of failures. Whereas the likeliness or chance is the number of successes divided by total number (success + failures). Therefore, it is not appropriate saying ‘more likelihood’ instead it should be ‘more odds’. “33 percent more” – this interpretation of OR = 1.33 is a bit decisive. This 33% is not the increase in proportion of PNC in the sample rather it is the percentage change of percentage. For example, if non-working mothers receive a PNC of proportion p1 and working mothers receive p2, then this 33% extracted from OR does not mean p2-p1 rather it means (p2-p1)/p1*100. Therefore, I would advise avoiding this sort of interpretation of OR. Keeping it simple for example ‘working mothers have 1.33 times more odds of having PNC.’

**Comment 3 Presenting numbers** – it would be great to see the numbers in table 1. As there is no other table showing the numbers in the article. The table title saying ‘percentage distribution of selected covariates’ however the percentage presented is in fact the distribution of PNC by the covariates.  Presenting both the columns for ‘yes’ and ‘no’ is redundant to me. As proportion of yes is equal to 1 minus the proportion of no.

Reviewers' comments (please check the attached file):

Reviewer's Responses to Questions

**Comments to the Author**

1. If the authors have adequately addressed your comments raised in a previous round of review and you feel that this manuscript is now acceptable for publication, you may indicate that here to bypass the “Comments to the Author” section, enter your conflict of interest statement in the “Confidential to Editor” section, and submit your "Accept" recommendation.

Reviewer #2: (No Response)

Reviewer #3: (No Response)

2. Is the manuscript technically sound, and do the data support the conclusions?

Reviewer #2: (No Response)

Reviewer #3: Yes

3. Has the statistical analysis been performed appropriately and rigorously? 

Reviewer #2: (No Response)

Reviewer #3: Yes

4. Have the authors made all data underlying the findings in their manuscript fully available?

Reviewer #2: (No Response)

Reviewer #3: Yes

5. Is the manuscript presented in an intelligible fashion and written in standard English?

Reviewer #2: (No Response)

Reviewer #3: No

6. Review Comments to the Author

Reviewer #2: This is an interesting paper which is carried our scientifically. Important results are emerged which have clear policy implications.

Reviewer #3: (No Response)

7. PLOS authors have the option to publish the peer review history of their article (what does this mean?). If published, this will include your full peer review and any attached files.

Reviewer #2: No

Reviewer #3: No

---

## [Author Response · Author response to Decision Letter 1]

9 Sep 2024

Authors responses to the review comments:

We would like to sincerely thank the anonymous reviewers, and the Academic Editor, for their valuable comments. We have considered all comments and then thoroughly revised and formatted the manuscript. A detailed response to each comment is provided in the tables as follows.

Author's response to Editor comments:

Thanks. We appreciate your feedback. As per comments, a careful revision has been conducted, and all required files are uploaded to the journal submission system. The revised texts are highlighted in “red” color.

Author's response to Journal Requirements:

Thanks. We checked the reference list and confirmed that it is complete and correct. 

The revised texts are in “red”. Page: 18-23

Author's response to Additional Editor Comments:

1. Thank you very much for your insightful comments and feedback. We believe that this helps to improve the quality of the manuscript. The authors used a secondary dataset and in the BDHS-2017/18 report, the PNC is defined. The PNC is for both mothers and newborns. 

2. Thanks. We appreciate your insightful comments. We have revised the manuscript accordingly. The revised texts are in “red”. Page:1, 12-13

3. Thanks. We revised the Table 1 as per your comments. The revised texts are in “red”. Page: 8-9

Author's response to Reviewer 3 Comments:

Thank you very much for carefully checking the manuscript and providing comments and feedback. We believe that this helps to improve the quality of the manuscript. 

Thanks. Basically, most of the previous studies used logistic regression to determine the contributing factors of PNC, however, the BDHS data have some between-cluster variation. In such a situation, multilevel analysis provides the precise calculation of regression coefficients and standard errors. Moreover, it provides more classification accuracy than the logistic regression model. The revised texts are in “red”. Page: 4

Thank you very much. We revised the Introduction section as per your review comments. The revised texts are in “red”. Page: 2-4

Thanks. We revised the equation and add the citations. The revised texts are in “red”. Page: 6

Thank you very much for your careful checking. We corrected it. The revised texts are in “red”. Page: 11

Thanks. To assess the discriminatory performance, the area under the ROC, i.e., AUC is discussed in Chapter 10 of your suggested book. We used this in our manuscript. The revised texts are in “red”. Page: 13-11

Thanks. We revised it. The revised texts are in “red”. Page: 2

We appreciate your suggestion. We revised the texts. The revised texts are in “red”. Page: 6

Thanks. We revised the manuscript as per comments and feedback. The revised texts are in “red”. Page: 1-16

In conclusion, the revised version of the manuscript has been produced as per the review outcomes. So, we hope that you will be happy to see this greatly improved version. Once again, we would like to thank you all for your dedication, professional services, and cooperation.

---

## [Decision Letter · Decision Letter 2]

24 Oct 2024

Determining the Influential Factors of Postnatal Care in Bangladesh using Multilevel Logistic Regression

PONE-D-23-08999R2

Dear Dr. Hossain,

We’re pleased to inform you that your manuscript has been judged scientifically suitable for publication and will be formally accepted for publication once it meets all outstanding technical requirements.

Kind regards,

Mohammed Moinuddin, PhD

Academic Editor

PLOS ONE

Additional Editor Comments (optional):

Reviewers' comments:

Reviewer's Responses to Questions

**Comments to the Author**

1. If the authors have adequately addressed your comments raised in a previous round of review and you feel that this manuscript is now acceptable for publication, you may indicate that here to bypass the “Comments to the Author” section, enter your conflict of interest statement in the “Confidential to Editor” section, and submit your "Accept" recommendation.

Reviewer #3: (No Response)

2. Is the manuscript technically sound, and do the data support the conclusions?

Reviewer #3: (No Response)

3. Has the statistical analysis been performed appropriately and rigorously? 

Reviewer #3: (No Response)

4. Have the authors made all data underlying the findings in their manuscript fully available?

Reviewer #3: (No Response)

5. Is the manuscript presented in an intelligible fashion and written in standard English?

Reviewer #3: (No Response)

6. Review Comments to the Author

Reviewer #3: (No Response)

7. PLOS authors have the option to publish the peer review history of their article (what does this mean?). If published, this will include your full peer review and any attached files.

Reviewer #3: No

---

## [Editor Report · Acceptance letter]

28 Oct 2024

PONE-D-23-08999R2 

PLOS ONE

Dear Dr. Hossain, 

I'm pleased to inform you that your manuscript has been deemed suitable for publication in PLOS ONE. Congratulations! Your manuscript is now being handed over to our production team.

Kind regards, 

on behalf of

Dr Mohammed Moinuddin 

Academic Editor

PLOS ONE